# Time Trend and Association of Early-Onset Colorectal Cancer with Diverticular Disease in the United States: 2010–2021

**DOI:** 10.3390/cancers14194948

**Published:** 2022-10-09

**Authors:** Lindsey Wang, Rong Xu, David C. Kaelber, Nathan A. Berger

**Affiliations:** 1Center for Science, Health, and Society, Case Western Reserve University School of Medicine, Cleveland, OH 44106, USA; 2Center for Artificial Intelligence in Drug Discovery, Case Western Reserve University School of Medicine, Cleveland, OH 44106, USA; 3Case Comprehensive Cancer Center, Case Western Reserve University School of Medicine, Cleveland, OH 44106, USA; 4The Center for Clinical Informatics Research and Education, The MetroHealth System, Cleveland, OH 44109, USA

**Keywords:** early-onset colorectal cancer, diverticular disease, electronic health records

## Abstract

**Simple Summary:**

The incidence of early-onset colorectal cancer (EOCRC; in patients <50 years old) has increased at an alarming rate. Diverticular disease is a common condition that can lead to inflammation of the colon. EOCRC and diverticular disease share many risk factors and are regarded as diseases of western civilization. We aimed to examine time trends of incidence rates of EOCRC from 2010 to 2021 among patients with and without preexisting diverticular disease and investigated whether diverticular disease is associated with increased risk of EOCRC. We show that the incidence of diverticular disease continuously increased from 2010 through 2021, and that diverticular disease is a significant risk factor for EOCRC. Our findings call for careful monitoring of EOCRC among patients with pre-existing diverticular disease and for more mechanistic research of the observed associations.

**Abstract:**

Purpose: To examine time trends of incidence rates of EOCRC from 2010 to 2021 among patients with and without diverticular disease and to examine whether diverticular disease is associated with increased risk of EOCRC. Methods: This is a retrospective cohort study of 46,179,351 young adults aged 20–49, including 298,117 with diverticular disease. We examined yearly incidence rate of first diagnosis of EOCRC from 2010 through 2021 among patients with and without diverticular disease. The 5-year risk of EOCRC among patients with pre-existing diverticular disease was compared to propensity-matched patients without diverticular disease and EOCRC and odds ratio (OR) and 95% confidence interval (CI) were calculated. Results: The yearly incidence rate of new diagnosis of EOCRC (measured as new cases per 100,000 people per year) in young adults with pre-existing diverticular disease increased from 100 in 2010 to 402 in 2021, 4–6 times higher than in those without diverticular disease (24 in 2010 to 77 in 2021) (*p* < 0.001). Patients with diverticular disease were at higher risk for EOCRC than those without (OR: 1.76, 95% CI: 1.40–2.32). Conclusion: The incidence of EOCRC continuously increased from 2010 through 2021 in patients with and without diverticular disease and was 4–6 times higher among patients with diverticular disease. Patients with pre-existing diverticular disease were at a significantly increased risk for EOCRC.

## 1. Introduction

The incidence of early-onset colorectal cancer (EOCRC; in patients <50 years old) has increased at an alarming rate [1,2,3]. Interestingly, while EOCRC has been reported to be increasing, the overall age-adjusted incidence rate of CRC decreased by 0.92% between 1975 and 2010 [4]. Multiple factors have been linked to EOCRC, including inflammatory bowel diseases, obesity, diet, sedentary lifestyle, alcohol consumption, smoking and mutations causing multiple cancer and predisposing syndromes such as hereditary nonpolyposis colon cancer (HNPCC), familial adenomatous polyposis (FAP), and familial CRC [3]. Diverticular disease is a common condition that can lead to inflammation, perforation and bleeding of the colon [5]. EOCRC and diverticular disease share many risk factors including obesity, smoking, diet (insufficient fiber, high red meat and fat) [3,6], and both are regarded as diseases of western civilization [3,7]. However, it remains unknown whether diverticular disease is associated with increased risk of EOCRC. Based on a nation-wide database of patient electronic health records, we examined time trends of incidence rates of first diagnosis of EOCRC from 2010 to 2021 among patients with and without preexisting diverticular disease and investigated whether diverticular disease is associated with increased risk of EOCRC by comparing propensity-score-matched cohorts with and without diverticular disease.

## 2. Materials and Methods

### 2.1. Database Description

We used the TriNetX Analytics network platform, which contains nation-wide and real-time (updated daily) de-identified electronic health records (EHRs) of 100 million unique patients from 68 health care organizations, mainly large academic medical institutions with both inpatient and outpatient facilities at multiple locations across all 50 states in the US [8], covering diverse geographic locations, age groups, racial and ethnic groups, income levels and insurance types. Though the data are de-identified, the built-in statistical functions within the TriNetX Analytics Platform can be used to perform statistical analyses on patient-level data. TriNetX reports population-level data and results without revealing protected health information (PHI) identifiers, so the MetroHealth System, Cleveland OH, Institutional Review Board has determined that any research using TriNetX is not Human Subject Research and is therefore exempt from review. We previously used the TriNetX Analytics network platform to conduct retrospective cohort studies [9,10,11,12,13,14,15,16,17], including examining COVID-19 risks and outcomes in patients with cancer [13,14,15].

### 2.2. Study Population

To examine the incidence rates of EOCRC between 2010 and 2021, the study population consisted of 46,179,351 young adults aged 20–49 who had a medical visit with healthcare organizations during the period 2010–2021, including 298,117 with preexisting diverticular disease and 45,881,234 without preexisting diverticular disease. As the incidence rate of EOCRC has been increasing over the years, we separately examined the associations of diverticular disease with EOCRC by calendar year. To examine whether diverticular disease is associated with increased risk for EOCRC, we used seven study populations by year (2010, 2011, 2012, 2013, 2014, 2015, 2016) that consisted of patients aged 20–44 (to allow 5-year follow-up before 50). Each study population for a given year (e.g., the 2016 study population) was then divided into two cohorts: (1) Diverticular disease (+) cohort—those who had a diagnosis of diverticular disease in that year; (b) Diverticular disease (−) cohort—those who did not have diverticular disease but had medical visits with healthcare organizations in that year. The status of diverticular disease was based on the International Classification of Diseases (ICD-10) diagnosis code K57 “Diverticular disease of intestine”. The status of CRC was based on ICD-19 diagnosis code C18 “Malignant neoplasm of colon”, C19 “Malignant neoplasm of rectosigmoid junction”, or C20 “Malignant neoplasm of rectum”.

### 2.3. Statistical Analysis

(1) We examined the incidence rate of first-time diagnosis of diverticular disease (measured by new cases per 100,000 persons per year) from 2010 through 2021 among patients age 20–49 who had a medical visit with a healthcare organization. We also examined the incidence rate of first-time diagnosis of EOCRC (measured by new cases per 100,000 persons per year) from 2010 through 2021 among patients aged 20–49 who had diverticular disease and among those who had no diverticular disease. Chi-squared test was performed to examine whether there was a linear trend in the incidence rates from 2010 through 2021, with significance set at *p*-value < 0.05 (two-sided).

(2) We examined whether patients in the Diverticular disease (+) cohorts had increased risk for EOCRC compared to those in the Diverticular disease (−) cohorts. The Diverticular disease (+) and Diverticular disease (−) cohorts from each of the 7 study populations were propensity-score matched (1:1 using a nearest neighbor greedy matching with a caliper of 0.25 times the standard deviation) for potential confounders. Propensity-score matching is a commonly used method for the analysis of non-randomized controlled trials to adjust for known confounding factors [18,19]. In this study, the following covariables were propensity-score matched between the Diverticular disease (+) and Diverticular disease (−) cohorts, in addition to demographics (age, gender, race/ethnicity): adverse socioeconomic determinants of health and life styles (e.g., problems related to socioeconomic and psychosocial circumstances, problems related to lifestyle, encounters for dietary counseling and surveillance, lack of physical exercise), alcohol drinking, tobacco smoking, overweight and obesity, personal history of malignant neoplasms of digestive organs, and medications including non-steroidal anti-inflammatory drugs (NSAIDs) and aspirin. All these factors are potential confounders, as they are related to risks of both diverticular disease and EOCRC [3,6].

(3) First-time diagnosis of EOCRC in matched cohorts were followed for 5 years. Odds ratio (OR) and 95% confidence intervals were calculated by comparing 5-year risk of EOCRC between propensity-score-matched Diverticular disease (+) and Diverticular disease (−) cohorts. For example, patients in the 2016 population (age 20–44) were followed from 2016 through 2021, and 5-year risks for EOCRC were compared between matched cohorts.

All statistical tests were conducted within the TriNetX Advanced Analytics Platform. The TriNetX platform calculates ORs and associated CIs using R, version 3.2-3. *p*-values were calculated from a t-test for continuous variable and Z-test for categorical variables with significance set at *p*-value < 0.05 (two-sided).

## 3. Results

### 3.1. Yearly Incidence Rate of Diverticular Disease and EOCRC among Young Adults from 2010 through 2021

The yearly incidence rate of diverticular disease among patients aged 20–49 continuously increased from 160 in 2010 to 406 in 2021 (*p* < 0.001). The yearly incidence rate was higher in men than in women (191 vs. 139 in 2020, and 486 vs. 353 in 2021) (Figure 1). The yearly incidence rate of new diagnosis of EOCRC among patients aged 20–49 who had no pre-existing diverticular disease increased from 2010 through 2020 (24 in 2010, 36 in 2015, 247 in 2020) and then decreased to 77 in 2021 (trend test: *p* < 0.001) (Figure 2). These results demonstrate that the increasing trend of EOCRC incidence rate previously reported from 1975–2010 [4] continued to increase until 2020. The incidence rate of EOCRC was higher in young adults aged 20–49 who had a diagnosis of diverticular disease than in those without pre-existing diverticular disease, and it significantly increased from 2010 through 2020 (100 in 2010, 165 in 2015, and 1393 in 2020) and decreased to 402 in 2021 (trend test: *p* < 0.001). More details and data are in the Appendix A). In summary, the incidence rates of both diverticular disease and EOCRC increased in young adults aged 20–49 during 2010–2021. In addition, the incidence rate of EOCRC was significantly higher in young adults with diverticular disease than in those without, and it significantly increased from 2010 through 2021. These results suggest that diverticular disease might be associated with increased risk of EOCRC.

### 3.2. Increased Risk of EOCRC among Patients with Pre-Existing Diverticular Disease Compared to Propensity-Score-Matched Patients without Diverticular Disease

Before propensity-score matching, the Diverticular disease (+) and Diverticular disease (−) cohorts differed significantly with respect to many of the potential confounders, including demographics (age, gender, race, ethnicity), adverse socioeconomic determinants of health and lifestyles, comorbidities, and medications that are relevant to both diverticular disease and EOCRC, as demonstrated by the 2016 population. Compared to the Diverticular disease (−) cohort, the Diverticular disease (+) cohort was older, comprised more male, Hispanic and White patients, had higher proportions of patients who had problems related to lifestyle, were users of tobacco, alcohol drinkers, NSAIDs or aspirin, and had higher prevalence of obesity and overweight (Table 1). After propensity-score matching, the Diverticular disease (+) and Diverticular disease (−) cohorts were balanced, demonstrating that effects of potential confounders were minimized by propensity-score matching (Table 1). Similar characteristics for other study populations were observed.

We compared 5-year risk of EOCRC (risk of developing EOCRC within 5 years after a diagnosis of diverticular disease) between propensity-score-matched Diverticular disease (+) and Diverticular disease (−) cohorts for seven populations (age 20–44). For each year’s population, the 5-year risk of first-time diagnosis of EOCRC was calculated and compared by following patients for 5 years in matched diverticular disease (+) and diverticular disease (−) cohorts. Compared to propensity-score-matched patients without diverticular disease, patients with pre-existing diverticular disease were at higher risk for first-time diagnosis of EOCRC within 5-years for all seven study populations. For example, for the 2016 study population, the 5-year risk for EOCRC was 1.21% among patients with diverticular disease, higher than the 0.69% among propensity-score matched patients without diverticular disease (HR: 1.76, 95% CI: 1.40–2.32) (Figure 3).

## 4. Discussion

We show that the incidence rate of diverticular disease among young adults continuously increased from 2010 through 2021. Risk factors for diverticular disease include age, diet (low fiber and high red meat and fat), smoking, alcohol drinking, obesity, lack of physical activity, and medications including NSAIDs and aspirin [6]. The incidence of diverticular disease increases with age, with greatest risk among people over 70 years old [6]. In this study, we show that the incidence of diverticular disease increased over time for young adults aged 20–49. The increasing incidence of diverticular disease is consistent with the trend of worsening diet [20], increasing incidence of obesity [21], and large increase in the use of aspirin and NSAIDs [22], though smoking and alcohol drinking have decreased over time in the US [23,24]. The incidence rate of diverticular disease was higher in men than in women, which is consistent with a previous report that women have lower odds of developing diverticular disease than men [25]. To the best of our knowledge, our study is among the first reporting the long-term (12-year) time trend of diverticular disease among young adults. Future research is necessary to identify risk factors that may have driven this continuous and substantial increase in diverticular disease among young adults in the US.

The yearly incidence rate of new diagnosis of EOCRC among young adults aged 20–49 (with or without diverticular disease) increased from 2010 through 2021, which is consistent with previous findings of rising EOCRC in the US from 1975 to 2017 [1,2,26,27]. There was a spike in the incidence rate of EOCRC in 2018 compared with that in 2017 or 2019. In 2018, the American Cancer Society guideline for colorectal cancer screening recommended that regular screening among average-risk adults should be lowered to begin at age of 45 years old [28]. The spike in 2018 suggests that the implementation of the new guideline may have picked up EOCRC cases that would not have been become clinically apparent until subsequent years. Our study also included data from two COVID-19 pandemic years. The incidence rate of EOCRC was 247 and 77 new cases per 100,000 people in 2020 and 2021, respectively, both of which are higher than the 58 reported in 2019. The reason for the peak in 2020 followed by a drop in new diagnoses of EOCRC in 2021 is unknown, but might be related to early diagnosis following severe COVID-19 and associated hospitalization and delayed cancer screening as a result of the COVID-19 pandemic [29]. Our previous studies on the early pandemic in 2020 showed that patients with cancer, including colorectal cancer, were more susceptible to SARS-CoV-2 infections and associated hospitalization due to their compromised immune functions [30,31], potentially leading to cancer diagnosis in patients who had cancer but had not yet been diagnosed. However, future research is needed to understand factors driving the continuous rise of EOCRC among young adults.

Our study is among the first to report the time trend of the annual incidence rate of EOCRC among young adults with diverticular disease, which increased from 2010 through 2021 and is 4–6 times higher than the rate among those without diverticular disease during the same time period. While these results suggest that diverticular disease might be associated with increased risk of EOCRC, young adults with diverticular disease differed significantly from those without diverticular disease: they were older, included more men and Hispanics, and showed a higher prevalence of adverse socioeconomic determinants of health and lifestyles, obesity, and use of NSAIDs and aspirin, all of which are risk factors for EOCRC. After propensity-score matching, young adults with pre-existing diverticular disease had a higher risk for EOCRC than those without. Previous studies have reported associations of diverticular disease with left-side colon cancer among patients who were mostly older than 50 years of age [32,33,34]. In our study, the status of CRC was based on ICD-10 diagnosis code C18 “Malignant neoplasm of colon”, C19 “Malignant neoplasm of rectosigmoid junction”, or C20 “Malignant neoplasm of rectum”, among which 67% were colon cancer. We did not separately examine the associations of rectal cancer with diverticular disease due to the limited sample size for each study population. As seen in Figure 1 and Figure 3, the incidence and number of young adults diagnosed with diverticular disease, as well as the incidence of EOCRC, have increased over years, which will provide sufficient samples in the future to examine the association of diverticular disease with different types of CRC including left-side rectal cancer. In addition, due to the limitations of sample size, we focused on diverticular disease, including both diverticulosis and diverticulitis, where the latter is a more severe form of diverticular disease. Future research is needed to examine whether the risk for EOCRC differs between patients with diverticulitis and those with diverticulosis.

This study shows a robust and significant association of diverticular disease with the risk of EOCRC. In this study, we separately examined the associations of diverticular disease with EOCRC in seven study populations by calendar year, which allowed comparison of matched Diverticular disease (+) and Diverticular disease (−) cohorts from the same year while minimizing confounding effects associated with calendar year. We observed consistently higher 5-year risk for EOCRC among patients with diverticular disease than in matched patients without diverticular disease for all seven study populations, though the strength of associations varied by calendar year. These results further support the robustness of our finding of significant associations of diverticular disease with EOCRC. Patients with diverticular disease are more likely to have a colonoscopy, which can lead to early detection of CRC. While this may lead to increased risk for CRC diagnosis in the short term, it did not have impact on long-term (e.g., 12-month) risk comparison, as shown in a previous study [34]. In this study, we compared 5-year risk for EOCRC between matched cohorts; therefore, biases in CRC screening should have been mitigated.

Inherited cancer predisposition, including hereditary syndromes (e.g., Lynch syndrome, familial adenomatous polyposis, Peutz-Jeghers syndrome, MUTYH-associated polyposis), and associated genes (e.g., MLH1, MSH2, MSH6, RPS20, APC, PTEN), accounts for about 13% of EOCRC [35]. Modifiable risk factors such as obesity, diet, sedentary lifestyle, and dysbiosis of the gut microbiota may account for a substantial portion of the risk for EOCRC, as well as the increasing incidence [3]. The findings in this study suggest that diverticular disease is a potential modifiable risk factor for EOCRC. Both incidence rates of diverticular disease and EOCRC among young adults have increased over the years, which alone does not suggest association, as it may be due to changes in diet patterns, the rise in obesity, and increased use of aspirin and NSAIDs. After matching for potential confounders including age, race and ethnicity, obesity, diet patterns, smoking, lifestyles, and medication uses, we observed consistent associations of diverticular disease with the risk of EOCRC. Future research is need to further understand the mechanisms underlying the observed association, which may including the role of inflammation and dysbiosis of the gut microbiota [36].

Our study has several limitations: First, this is a retrospective observational study, so no causal inferences can be drawn. Second, there are inherent limitations in studies based on patient EHRs, including over-/mis-/underdiagnosis and unmeasured confounders. However, both Diverticular disease (+) and Diverticular disease (−) cohorts were drawn from the same standardized TriNetX database of electronic health records, so these issues should not substantially affect the relative risk comparison. Third, while the propensity-score matching method adjusted for known confounding factors that have been documented in patient electronic health records, observational studies including this EHR-based cohort study are prone to unmeasured or unknown confounding. However, fully controlling for unknown confounding factors can be achieved only in randomized controlled trials [19]. Fourth, patients in the TriNetX database represented those who had medical encounters with the healthcare systems contributing to the TriNetX Platform. Even though this platform includes 28% of the US population, it does not necessarily represent the entire US population. Therefore, the results from the TriNetX platform need to be validated in other populations.

## 5. Conclusions

In summary, young adults with diverticular disease were at increased risk for EOCRC. Future studies are necessary to understanding mechanisms underlying the observed association of diverticular disease with EOCRC.

## Figures and Tables

**Figure 1 cancers-14-04948-f001:**
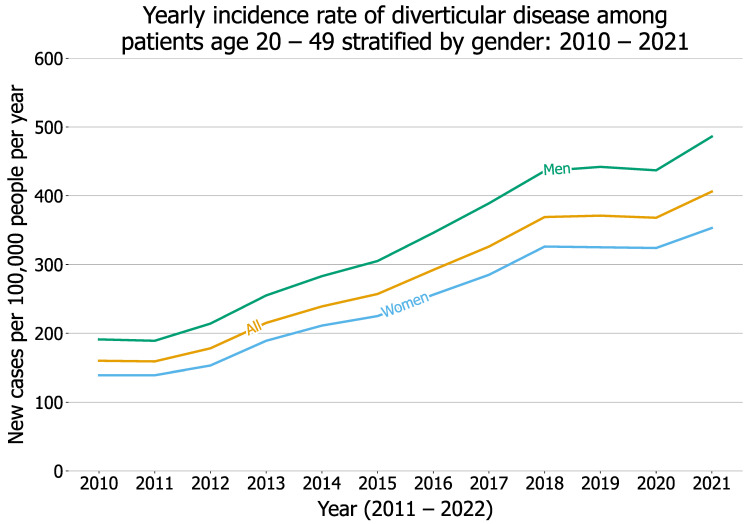
Yearly incidence rate of diverticular disease (measured by new cases per 100,000 people per year) between 2010 and 2021 among young adults aged 20–49.

**Figure 2 cancers-14-04948-f002:**
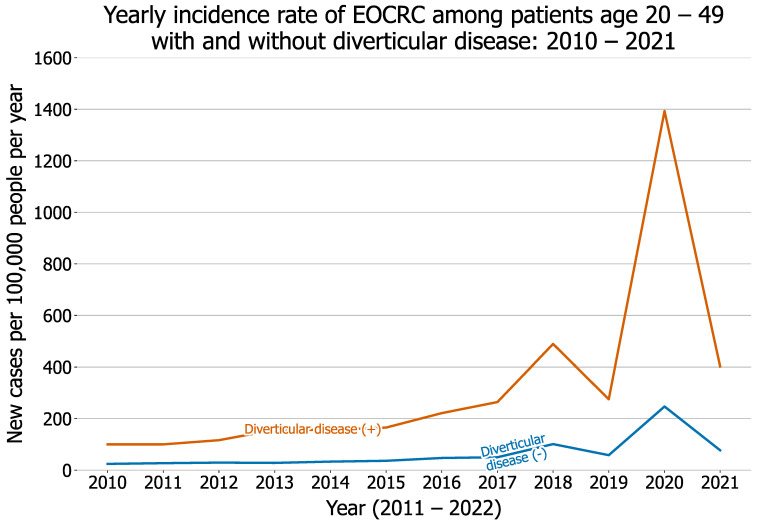
Yearly incidence rate of EOCRC (measured by new cases per 100,000 people per year) between 2010 and 2021 among young adults (age 20–49) with and without diverticular disease.

**Figure 3 cancers-14-04948-f003:**
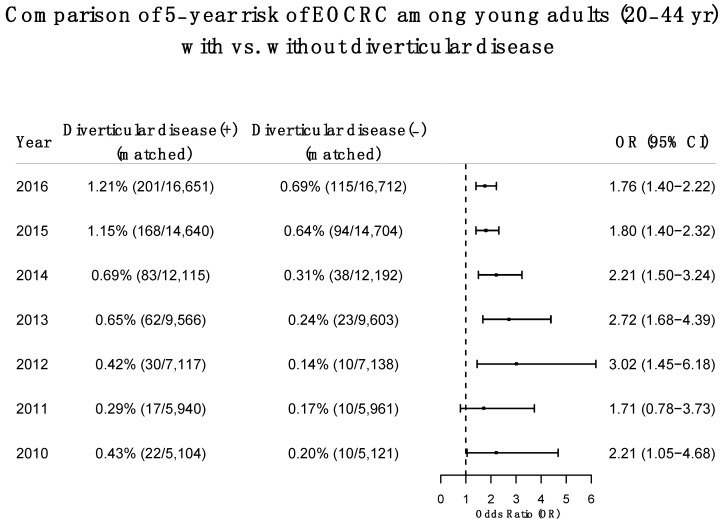
Comparison of 5-year risk of EOCRC in propensity-score-matched Diverticular disease (+) and Diverticular disease (−) cohorts for 7 study populations (age 20–44). Cohorts were propensity-score matched for potential confounders including demographics (age, gender, race, ethnicity), adverse socioeconomic determinants of health and lifestyles, comorbidities, and medications. First-time diagnosis of EOCRC was followed for 5 years, for example from 2016 after the diagnosis of diverticular disease through 2021 for the 2016 study population.

**Table 1 cancers-14-04948-t001:** Characteristics of the 2016 study population before and after propensity-score matching. Diverticular disease (+)—20–44 years old in 2016 and had a diagnosis of diverticular disease in 2016; Diverticular disease (−) cohort—20–44 years old in 2016, never had a diagnosis of diverticular disease but had medical encounters with healthcare organizations in 2016. SMD—standardized mean differences. * SMD greater than 0.1, a threshold being recommended for declaring imbalance. Self-identified race and ethnicity as recorded in the TriNetX database were included because they have been associated with both diverticular disease risk and EOCRC.

	Before Matching			After Matching		
	Diverticular Disease (+) Cohort (Year 2016)	Diverticular Disease (−) Cohort (Year 2016)	SMD	Diverticular Disease (+) Cohort (Year 2016)	Diverticular Disease (−) Cohort (Year 2016)	SMD
**Total number**	16,782	2,991,298		16,782	16,782	
**Age at index event (years, mean ± SD)**	37.3 ± 5.7	32.4 ± 7.1	0.76 *	37.3 ± 5.7	37.2 ± 5.7	0.003
**Sex (%)**						
Female	48.3	65.5	0.35 *	48.3	48.0	0.005
Male	51.7	34.5	0.35 *	51.7	52.0	0.005
**Ethnicity (%)**						
Hispanic/Latinx	16.9	9.3	0.23 *	16.9	17.0	0.002
Not Hispanic/Latinx	61.8	49.8	0.24 *	61.8	61.8	0.001
Unknown	21.3	40.9	0.43 *	21.3	21.2	0.003
**Race (%)**						
Asian	1.0	2.3	0.09	1.0	1.1	0.003
Black	14.4	13.5	0.03	14.4	14.2	0.005
White	69.2	52.3	0.35 *	69.2	70.0	0.02
Unknown	14.4	31.4	0.41 *	14.4	13.8	0.02
**Adverse socioeconomic determinants of health and life styles (%)**						
Health hazards related to socioeconomic and psychosocial circumstances	3.1	2.1	0.06	3.1	2.8	0.02
Problems related to lifestyle	4.7	2.3	0.13 *	4.7	4.2	0.03
Dietary counseling and surveillance	1.4	0.8	0.06	1.4	1.2	0.02
Lack of physical exercise	0.1	0.03	0.02	0.1	0.1	<0.001
Alcohol drinking	4.5	1.9	0.15 *	4.5	3.9	0.03
Tobacco smoking	14.8	6.5	0.27 *	14.8	14.7	0.003
**Pre-existing medical conditions (%)**						
Overweight and obesity	19.6	8.4	0.33 *	19.6	19.3	0.009
Personal history of malignant neoplasm of digestive organs	0.4	0.1	0.08	0.4	0.4	0.008
**Medications (%)**						
NSAIDs	23.4	16.4	0.17 *	23.4	23.0	0.008
Aspirin	8.9	3.7	0.22 *	8.9	8.1	0.03

## Data Availability

The data can be shared up on request.

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
