# Peer review of "Time Trend and Association of Early-Onset Colorectal Cancer with Diverticular Disease in the United States: 2010–2021"

_cancers, 2022, doi:10.3390/cancers14194948_

Round 1

Reviewer 1 Report (Previous Reviewer 2)

The authors present a retrospective cohort study of the incidence of early onset colorectal cancer (EOCRC) and diverticulosis based on an electronic query of a US national patient database. The data include more than 46 million patients aged 20 to 49 years, of whom about 300000 had diverticulosis. Two propensity-matched cohorts each with and without preexisting diverticulosis were studied from 2010 to 2021. Several external risk factors were taken into account. The results show a 4-6-fold higher incidence of EOCRC in the cohort with preexisting diverticulosis and a continuous increase in incidence since 2010, and the authors conclude that the results suggest an association between diverticulosis and EOCRC.
The text is clearly organized. The tables and figures in the manuscript and the supplements are easy to understand. The strengths of the manuscript are primarily large amount of data and the remarkable result of an association of EOCRC and diverticulosis.

I notice shortcomings in the content that should be addressed before publication:

Line 34 ff: the explanation of incidence should not be repeated five times in the text (ll. 34, 95, 127, 131, 146). If anything, the execution can remain once in l. 95. Rather, the introduction should briefly discuss the methodology of propensity score matching in cohort studies and outline its strengths and weaknesses compared with randomized case-control studies.

Line 113: Reducing genetically defined tumors to one of many confounding factors does not do justice to their importance in the pathogenesis of EOCRC. Rather, these mutations represent a direct causal genesis of up to 50% of observed tumor disease, from which targets for therapies are also derived. It is highly doubtful that there is a causal relationship with possible concomitant diverticula in these cases, which would represent a systematic error. To estimate the propensity score, it is essential to include such causal factors as confounders. Please comment on why this was not done in your approach. Was this information impossible to collect in the TriNetX database? How do they ensure that this essential item is equally distributed in both matched cohorts? This point should definitely be worked out in the discussion.

Line 223-225: Cohort studies are considered essential epidemiologic tools for detecting pathogenetic associations. Correlation does not imply causality, however, many readers do not distinguish this consistently!

Figure 2: How do you explain the decline in incidence in 2019? Are there possibly effects of delayed reporting until 2020?

Line 244-246: At this point it should be stated that this study shows a remarkable coincidence, but does not allow conclusions on a causal relationship. Possible other common risk factors that have not been considered as confounders in this study so far, e.g., the microbiome (Hofseth et al., Nat Rev Gastroenterol Hepatol 2020), could be discussed here.
Line 270-272: As stated in the previous sections, the data show an association of diverticula and EOCRC. However, in my opinion, the data do not justify the conclusion to speak of a risk factor. Please reconsider the wording.

Furthermore, I notice a minor formatting error in the heading of chapter 2.3.

Round 2

Reviewer 1 Report (Previous Reviewer 2)

This retrospective cohort study examined the association of early onset colorectal cancer (EOCRC) with diverticular disease on a large database. The focus was on whether diverticulosis was associated with a higher risk of EOCRC. Propensity matching was used to ensure equal ratios for various confounding variables in cohorts with and without preexisting diverticular disease. Results showed increasing incidence of both conditions and an association of EOCRC with diverticulosis.

The results are noteworthy and may lead some clinical colleagues to reconsider the place of diverticulosis in carcinogenesis. My caveats regarding the role of hereditary tumor syndromes in the patient population with EORC were adequately addressed. Since only a part of EOCRC is associated with hereditary tumor syndromes, it can be assumed that the predominant part is associated with possibly modifiable risk factors.

Even if no causal relationship between EOCRC and diverticulosis can be established so far, studies like this one represent valuable basic building blocks for further etiological research of colorectal carcinomas. I consider these results relevant to a broader readership and therefore advocate the publication of this study in a prestigious journal such as "Cancers."

This manuscript is a resubmission of an earlier submission. The following is a list of the peer review reports and author responses from that submission.

Round 1

Reviewer 1 Report

The main aim of this work is to investigate the relationship between diverticular disease and colorectal cancer in the subject from a retrospective series of over 46 million adults under 49 years of age from the US Database. The incidence of CRC in this age group doubled between 2010 and 2021 (160 vs 406/100,000 persons/year). The incidence of CRC appears to be higher in people under 49 years of age with diverticular disease. The analysis of factors that may influence the occurrence of colonic cancer in subjects under 49 years of age is comprehensive, and includes several factors including obesity. Obesity is a risk factor for diverticular disease and a risk factor for colorectal cancer in young people. The authors recall this in the article by Akimoto et al, Rising incidence of early-onset colorectal cancer - a call to action.

Akimoto N, Ugai T, Zhong R, Hamada T, Fujiyoshi K, Giannakis M, Wu K, Cao Y, Ng K, Ogino S.

Nat Rev Clin Oncol. 2021.

Causality and correlation should not be confused. The work of Wang et al shows a correlation between diverticular disease and colorectal cancer in young people, but not causality. The conclusion must therefore be more nuanced. It would be regrettable if it led to abusive colectomies.

Reviewer 2 Report

The authors present a retrospective cohort study of the incidences of early-onset colorectal cancer (EOCRC) and diverticulosis based on an electronic query of a US national patient database. More than 46 million patients aged 20-49 years were included, of which approximately 300000 had diverticulosis. Annual incidences for EOCRC were also determined and statistically analyzed. Risks for EOCRC were compared for patients with and without concomitant diverticulosis in propensity-matched cohorts. A number of external risk factors for the development of colorectal cancer and diverticula were considered. The data showed an increasing incidence of EOCRC and diverticulosis over the years and a higher risk of EOCRC in the cohort of patients with concomitant diverticulosis. The authors conclude that the results suggest an association between diverticulosis disease and EOCRC.

The text is clearly structured and easy to read. The tables and figures are easy to understand. For the classification of the results, quite predominantly the latest literature is used; the oldest reference on the association of CRC and diverticulosis dates from 2011.

However, the study has a significant flaw that, in my opinion, precludes publication in this form. Known facts about EOCRC are not included in this study design.

From data previously published in 2015 over an observation interval of 1975-2010 (Bailey et al, JAMA 2015), it is apparent that the incidence of EOCRC is increasing at 2% per year, while the incidence of CRC is decreasing in patients >50y. It is by no means that this trend is observed for the first time in the present study. More serious, however, is the fact that already accepted molecular findings were not considered in the study design. 50% of EOCRC have a known cause (30% of patients with hereditary tumor syndromes, 20% with familial CRC), which should be considered independent of diverticulosis (Mauri et al, Molecular Oncology 2018). These molecularly defined entities are not considered or even mentioned in the present study in any way, even the search function does not provide any hits for the search terms "hereditary", "familial", "FAP" or "HNPCC". I consider it to be a serious systematic error to draw conclusions on an association with diverticular disease without first considering and extracting the subgroup of patients with a defined molecular pathogenesis.

Furthermore, I find flaws of a formal nature. In the methodology section, the total number of patients (~46 million) and the total number of patients with diverticulosis (~300000) are given, but the total number of tumor patients was not considered. Instead, annual numbers and incidences are shown. Unfortunately, the abstract gives differing numerical values for 2016 and 2021 (100 vs. 160 in 2016 and 402 vs. 406 in 2021, respectively). A tabular presentation of the total numbers and the year-specific case numbers would make the data clearer and easier to understand.

Table 1 is likely to remain unclear to many readers who are not very familiar with the method of propensity scores. The results section should go into more detail about the visible or not visible effects for each variable.

Summary:

The present study has a very interesting dataset with large case numbers as a basis and already the absolute numbers on diverticulitis and early-onset CRC are interesting. From a practical point of view, it is currently assumed that diverticular disease and CRC are pathogenetically distinct processes that nevertheless can occur together and may cause diagnostic difficulties. In this respect, it is appreciated to statistically investigate this large patient cohort for a possible association with CRC and especially EOCRC, as the latter are only 50% etiologically clarified. However, this expected large number of cases with known molecular pathogenesis must be extracted from the cohort of EOCRC and/or considered separately.

I ask the authors to revise and to expand or clean up the data in this regard. In the years since 2016, it can be assumed that the patients were examined for the presence of a hereditary or familial tumor syndrome (in our department, this has been standard for years for patients < 50 years). If these data cannot be collected with the present platform, the suitability for this specific question has to be critically questioned. In this case, the validity of the study is extremely limited and this must be discussed accordingly by the authors. It would be very unfortunate if diagnostic and possibly therapeutic consequences were built on this deficient data.

In its current form, the paper does not seem suitable for a prestigious journal such as Cancers and I must advise the editors to reject the paper.

References:

Bailey CE, Hu CY, You YN, Bednarski BK, Rodriguez-Bigas MA, Skibber JM, Cantor SB, Chang GJ. Increasing disparities in the age-related incidences of colon and rectal cancers in the United States, 1975-2010. JAMA Surg. 2015 Jan;150(1):17-22. doi: 10.1001/jamasurg.2014.1756. Erratum in: JAMA Surg. 2015 Mar 1;150(3):277. PMID: 25372703; PMCID: PMC4666003.

Mauri G, Sartore-Bianchi A, Russo AG, Marsoni S, Bardelli A, Siena S. Early-onset colorectal cancer in young individuals. Mol Oncol. 2019 Feb;13(2):109-131. doi: 10.1002/1878-0261.12417. Epub 2018 Dec 22. PMID: 30520562; PMCID: PMC6360363.